# Evaluation of the Genetic Diversity, Population Structure and Selection Signatures of Three Native Chinese Pig Populations

**DOI:** 10.3390/ani13122010

**Published:** 2023-06-16

**Authors:** Ziqi Zhong, Ziyi Wang, Xinfeng Xie, Shuaishuai Tian, Feifan Wang, Qishan Wang, Shiheng Ni, Yuchun Pan, Qian Xiao

**Affiliations:** 1Hainan Key Laboratory of Tropical Animal Reproduction & Breeding and Epidemic Disease Research, College of Animal Science and Technology, Hainan University, Haikou 570228, China; zhongziqi2021@163.com (Z.Z.); wff1195394551@163.com (F.W.); 2Hainan Yazhou Bay Seed Laboratory, Yongyou Industrial Park, Yazhou Bay Sci-Tech City, Sanya 572025, China; 3Department of Animal Science, College of Animal Science, Zhejiang University, Hangzhou 310058, China; 4Animal Husbandry Technology Extending Stations of Hainan Province, Haikou 570203, China

**Keywords:** Chinese indigenous pigs, single nucleotide polymorphism, genetic variation, conservation, F-statistic

## Abstract

**Simple Summary:**

Protection of genetic diversity is important for the sustainable development of the animal industry. Due to the unique environment in Hainan Province, the local pig populations have excellent heat resistance and adaptability. Moreover, the meat quality of these pigs is excellent and very popular with the market. Analysis of the genetic composition of the population may provide valuable information for conservation and utilization of the breed. Therefore, in our study, we aimed to assess the genetic diversity and population structure of three Chinese indigenous pig populations in Hainan Province, and further identified candidate selection signatures based on whole-genome sequencing data. Our analysis shows that native Chinese pig populations are more diverse than commercial pigs. In addition, analysis of selection signatures revealed some candidate genes related to certain traits, such as stress resistance.

**Abstract:**

Indigenous pig populations in Hainan Province live in tropical climate conditions and a relatively closed geographical environment, which has contributed to the formation of some excellent characteristics, such as heat tolerance, strong disease resistance and excellent meat quality. Over the past few decades, the number of these pig populations has decreased sharply, largely due to a decrease in growth rate and poor lean meat percentage. For effective conservation of these genetic resources (such as heat tolerance, meat quality and disease resistance), the whole-genome sequencing data of 78 individuals from 3 native Chinese pig populations, including Wuzhishan (WZS), Tunchang (TC) and Dingan (DA), were obtained using a 150 bp paired-end platform, and 25 individuals from two foreign breeds, including Landrace (LR) and Large White (LW), were downloaded from a public database. A total of 28,384,282 SNPs were identified, of which 27,134,233 SNPs were identified in native Chinese pig populations. Both genetic diversity statistics and linkage disequilibrium (LD) analysis indicated that indigenous pig populations displayed high genetic diversity. The result of population structure implied the uniqueness of each native Chinese pig population. The selection signatures were detected between indigenous pig populations and foreign breeds by using the population differentiation index (*F_ST_*) method. A total of 359 candidate genes were identified, and some genes may affect characteristics such as immunity (*IL-2*, *IL-21* and *ZFYVE16*), adaptability (*APBA1*), reproduction (*FGF2*, *RNF17*, *ADAD1* and *HIPK4*), meat quality (*ABCA1*, *ADIG*, *TLE4* and *IRX5*), and heat tolerance (*VPS13A*, *HSPA4*). Overall, the findings of this study will provide some valuable insights for the future breeding, conservation and utilization of these three Chinese indigenous pig populations.

## 1. Introduction

Domestication has led to the development of many new breeds of the common pig. In China, abundant pig populations have been formed, which can provide valuable germplasm resources for promoting the sustainable development of the pig industry around the world. Hainan Province, located in the most southern part of China and surrounded by the sea on all sides, has a tropical island climate characterized by warm and abundant rainfall throughout the year [1].The unique climate conditions and relatively closed geographical environment have led to the development of two high-quality pig breeds, namely, Hainan pigs and Wuzhishan (WZS) pigs. Hainan pigs can be further divided into four subpopulations: Tunchang pigs (TC), Dingan pigs (DA), Lingao pigs (LG) and Wenchang pigs (WC). These genetic resources perform well on many economic traits, such as stress resistance, disease resistance and juicy/tasty meat quality [2]. Additionally, these locally adapted tropical pig breeds are important sources of genes for thermotolerance—an invaluable conservation priority, particularly given the changing climate. Though not often consumed for meat, the Wuzishan pig is a miniature pig often used as a model organism for studying various human diseases [3].

However, decreases in annual lean meat production, poor growth rates and intense competition from exotic commercial pig breeds has led to a decline in the indigenous pig population. In recent years, although some measures and efforts have been implemented by the government to promote indigenous pig breeding, such as establishing national and provincial conservation farms for WZS, TC, DA and LG, we are still unsure if these conservation efforts have effectively protected the valuable genes expressed within the population. To date, the effect of this conservation effort has not been investigated based on molecular markers. Additionally, the population structure of these indigenous pig breeds has suffered tremendous shocks from African swine fever since 2018 [4]. Therefore, for better conservation of these genetic resources, population genetic analysis of the existing Hainan pig breeds is necessary.

Genetic diversity is the fundamental foundation for genetics and breeding research, and it can often represent the potential of a population to adapt to change [5]. Understandably, protection of biological genetic resources and diversity is one of the most important issues in the world. Another important metric of genetic analysis is population structure. This type of analysis can provide critical insights into potential genetic substructures in populations, serving as an important basis for any genetic data analysis [6]. Knowledge of the basics of the species’ genetics is essential to ensure accurate and sustainable management of its population. Similarly, understanding the genetic structure of a population is significant for comprehending individual migration, introgression, the evolution process of said population and breed classification. Unsurprisingly, there has been considerable research in animal species using genetic markers, including pigs [7,8,9]. For instance, Chen et al. (2022) investigated the population structure and genetic diversity of four indigenous pig breeds from eastern China based on single nucleotide polymorphism (SNP) chip and provided new insights for further conservation of these indigenous genetic resources [10]. However, despite this progress, the genetic diversity of the pig breeds in Hainan Province in China has rarely been reported.

Domestication may also leave ‘imprints’ in the sequence of the genome associated with distinct phenotypic characteristics. These imprints mainly manifest in the reduction of genetic diversity and increase in the degree of linkage disequilibrium [11]. In recent years, extensive research has identified many regions in the germplasm associated with positive traits, allowing us to identify selection signals indicating good breeding stock [12,13,14]. Thus, for systematic understanding of these three native Chinese pig populations, a comprehensive investigation of genetic diversity, genetic structure and selection signatures is necessary.

The objective of this present study is to investigate the genetic diversity and structure of three local pig populations in China, including Wuzhishan (WZS), Tunchang (TC) and Dingan (DA), based on molecular markers. Furthermore, we looked to identify potential candidate genes associated with high-quality economic characteristics known to the species (meat quality, heat tolerance and disease resistance). Our results will add to the growing knowledge on the genetic status and evolutionary history of these three indigenous pig populations and provide valuable insight into the genetic mechanisms of some phenotypic characters of growing interest in the changing climate such as environmental heat adaptation.

## 2. Materials and Methods

### 2.1. Sample Collection and Sequencing

A total of 30, 18 and 30 ear tissue samples of Wuzhishan (WZS), Tunchang (TC) and Dingan (DA) were collected from randomly chosen individuals (Table 1). The extraction of genomic DNA from the ear tissues was performed according to the procedure of a commercial kit (Tiangen Biotech Co., Ltd., Beijing, China). NanoDrop 2000 spectrophotometer (Thermo Scientific, Wilmington, DE, USA) was used to quantify the extracted DNA. The DNA was randomly fragmented through an ultrasonic high-performance sample processing system. After fragment selection, fragments of about 500 bp were obtained. Subsequently, the fragment ends were repaired, the 3’ end added with “A” base and both ends were added with a library adapter. The library tended to be DNA nanoball (DNB) after single-stranded separation, circularization and rolling circle amplification. After quality control, qualified libraries were allowed for sequencing.

The DNBSEQ platform (150 bp paired-end) was used for sequencing. The average sequenced depth was more than 15×. In addition, data from 13 individuals of Landrace (LR) and 12 individuals of Large White (LW) were acquired from the SRA database (https://www.ncbi.nlm.nih.gov/sra) (Appendix A). These data were used in this study to represent ‘commercial’ pigs and better contrast the genetic diversity of the indigenous pigs against a more typical population (Appendix A).

### 2.2. Quality Control and Genotyping

SOAPnuke (v2.1.0) was carried out for quality control (QC) of the reads [15], including elimination of the reads contaminated with adapter sequences, low-quality (more than 30% base with Phred quality less than 20) and reads containing more than 10% N bases. After filtration, reads from all individuals were aligned to the pig reference genome (Sscrofa11.1, https://www.ncbi.nlm.nih.gov/assembly/GCF_000003025.6/) (accessed on 1 December 2022) using the “MEM” algorithm of the BWA (v0.7.17) [16] alignment software. SNP detection was performed using the HaplotypeCaller tool of the GATK (v4.1.6.0) [17] software. High-quality SNPs, with max-missing less than 0.5 and minor allele frequency (MAF) more than 0.05, were kept with VCFtools (v0.1.16) [18] for the following analysis. Finally, ANNOVAR (version: 2020 Jun 07) [19] was used for the annotation and categorization of SNPs.

### 2.3. Genetic Diversity and Linkage Disequilibrium

Genetic diversity refers to the range of alleles and genotypes that exist within the studied population [20]. Various parameters are used to describe genetic diversity, which are useful indicators in population genetics. In this study, multiple indicators were employed to estimate the genetic diversity of indigenous pigs. VCFtools and Plink (v1.90) were performed to estimate the value of MAF, observed heterozygosity (H_O_) and expected heterozygosity (H_E_) [21], and *pi* values were calculated to evaluate nucleotide polymorphism. The genetic diversity parameters were statistically analyzed using violin plots generated with the ggplot2 package in R. The degree of linkage disequilibrium (LD) decay was measured by taking the squared correlation (*r^2^*) of pairwise SNPs using PopLDdecay with default parameters [22].

### 2.4. Genetic Differentiation Analysis

The F-statistic (*F_ST_*) is a statistical test used to measure the degree of population differentiation between groups [23]. In this study, we utilized VCFtools to evaluate *F_ST_*, resulting in a 5 × 5 *F_ST_* matrix that was visualized by heatmap. To obtain these results, 100 kb sliding windows were used with a 10 kb step size during the *F_ST_* calculation.

### 2.5. Population Genetic Structure

MEGA software (v7.0) [24] was carried out to construct a neighbor-joining (NJ) tree, which was based on the matrix data of identity-by-state distance. Principal component analysis (PCA) was performed by Plink (v1.90), and the PCA graph was plotted using the qqplot2 function in the R package. The ADMIXTURE v1.30 software [25] was employed to infer the proportion of introgressed ancestry in the tested populations. We calculated the values of k ranging from 2 to 4 and plotted them using the pophelper package in R.

### 2.6. Detection of the Selection Signatures

The *F_ST_* values between the indigenous pig populations and two exotic breeds were calculated using VCFtools. The analysis was performed across genomes using 100 kb windows and 10 kb steps. To empirically identify candidate regions undergoing positive selection, we have set the top 1% of *F_ST_* as the threshold to ensure data quality. To identify candidate genes in these regions, Ensembl database (Sscrofa 11.1; http://asia.ensembl.org/index.html) (accessed on 21 March 2023) was carried out [26].

### 2.7. Functional Enrichment Analysis of Candidate Genes

DAVID is a bioinformatics database which integrates biological data and analysis tools. The enrichment analysis of the candidate genes obtained in this study was performed by using DAVID with GO and KEGG pathway analyses [27]. Additionally, we downloaded published pig QTLs from the pig QTLdb database [28] to identify any overlap of the candidate regions. We compared the top 1% weighted values of *F_ST_* sliding windows with the pig QTL database to identify genes that may be associated with economically important traits.

## 3. Results

### 3.1. Single Nucleotide Polymorphism Discovery

A total of 28,384,282 SNPs were identified in these five pig populations. In the analysis, a total of 27,134,233 SNPs were identified from three types of indigenous pigs. ANNOVAR software was used to annotate the variation information, and a chart displaying the annotation information was plotted (Figure 1). Notably, the majority of SNPs were detected in the region of intergenic (44.45%) and introns (42.92%), with only a small percentage (0.67%) located in exonic regions. Appendix A displays the density distribution of SNPs across the chromosomes of the three indigenous pig types.

### 3.2. Analysis of Genetic Diversity and Linkage Disequilibrium

In order to estimate the genetic diversity of all pigs, five indices have been carried out (Table 1). We observed that the MAF of indigenous pigs was 0.22–0.23, with a standard deviation of approximately 0.13–0.14. For comparison, the MAF of commercial pigs was 0.24–0.26.

Heterozygosity is a commonly used parameter to measure genetic diversity. In this study, both observed heterozygosity (H_O_) and expected heterozygosity (H_E_) were used to estimate the genetic diversity of every tested pig population. The LR and WZS pig populations were found to have the lowest H_O_ and H_E_ values among all populations, while the LW population had the highest H_O_ (0.39) and H_E_ (0.35). Across all indigenous pig populations, the H_O_ values ranged from 0.29 to 0.34, while the H_E_ values ranged from 0.31 to 0.32.

Nucleotide diversity maps were generated for each pig population using the R package ggplot2 (Figure 2A). Interestingly, our analysis revealed that local pigs exhibit higher nucleotide diversity than commercial pig breeds. The WZS pig breed had the highest *pi* value, while the LW pig breed had the lowest *pi* value. Further analysis of the LD plot revealed that LW had the highest level of linkage disequilibrium, followed by LR. WZS had the lowest value, which is in line with the results of the nucleotide diversity plot (Figure 2B). The genetic dissimilarity between each pair of pig populations was investigated by using index of *F_ST_* value, with weighted *F_ST_* used for statistical analysis (Figure 2C). The genetic differentiation between WZS, TC and DA pig populations ranged from 0.14 to 0.15. The two commercial pig breeds, LW and LR, are relatively close at 0.18, while there is a significant gap between WZS, TC and DA pig populations and LW and LR pig breeds.

### 3.3. Analysis of Genetic Structure

We constructed a NJ tree to analyze the genetic distances between local and commercial pig breeds, which will help us to understand the phylogenetic relationship among these tested pigs (Figure 3A). We can clearly see in the NJ tree that the three types of indigenous pigs are mostly located at one end, while commercial pigs are mostly located at the other end and are far apart, which is consistent with the results of previous *F_ST_* pairwise analysis. The PCA analysis also clearly distinguished the commercial and local pig populations, indicating significant genetic differences between these two groups (Figure 3B). The first two PCAs explained only 24.68% and 6.41% of the total variation, respectively. However, despite this, there was obvious clustering. LR and LW clustered together, while the three indigenous pigs, WZS, TC and DA, clustered together. When we conducted a structural analysis and attempted to cluster the pig breeds into two groups, there was a clear grouping of the Chinese local pig populations and a distinct separate grouping of LR and LW. However, when we allowed the species to cluster into three groups, we observed that TC and DA formed one group of similar genetic profile, while WZS pigs were classified as independent individuals (Figure 3C). This might suggest that TC and DA breeds are more closely related.

### 3.4. Selection Signature and Functional Clustering of Variant Genes

We also compared the genomic signatures of indigenous pig populations (WZS, TC and DA) with commercial pig breeds (LR and LW) at the population level. We compared evidence of selection using *F_ST_* and annotated the obtained loci using the Ensembl database (http://asia.ensembl.org/index.html). By screening candidate genes using a 1% *F_ST_* value for statistics (weighted *F_ST_*; *F_ST_* > 0.536683), we found a total of 359 genes (Appendix A), shown in Figure 4. We performed enrichment analysis of these 359 genes using DAVID. After annotation, we found that 34 GO terms and 3 pathways were significantly enriched (*p* < 0.05) (Appendix A). Among the significant GO terms, Biological Process (BP) accounted for 38.24%, Cellular Component (CC) accounted for 32.35% and Molecular Function (MF) accounted for 29.41%. A bubble chart was drawn for all GO terms and pathways (Figure 5A–D).

A number of cellular components of metabolism were found to be upregulated in the GO analysis, including nucleus activity, myosin phosphatase activity, adaptive immune response, etc. The most significant GO term was associated with energy (interleukin-2 receptor binding, GO:0005134). In our KEGG pathway analysis, we discovered numerous signaling pathways that are potentially related to immune and metabolic processes. These findings may be associated with the excellent disease resistance and adaptability observed in Chinese indigenous pigs. As we delved further into our analysis, we compared the results with those of the Pig QTLdb database (Release 49, 28 December 2022). Through this comparison, a number of QTLs related to meat quality and health were identified (Appendix A), which may provide a deeper insight in understanding the genetic mechanism under the phenotypes of Hainan local pigs. Through our analysis of the GO term, KEGG pathways and comparison with the Pig QTLdb database, we identified several important candidate genes that may play a role in determining the genetic characteristics observed in Chinese indigenous pigs. These candidate genes include ABCA1, ADIG, TLE4 and IRX5. In addition, we have also found some genes related to immunity and adaptability, etc.

## 4. Discussion

### 4.1. Genetic Diversity and Linkage Disequilibrium Analysis of Chinese Indigenous Pigs and Commercial Pigs

Chinese native pigs are ancient compared to commercial pig breeds. Unlike Western pigs, Chinese pigs have not been subjected to the same level of intense selection pressure throughout history. As a consequence, Chinese pigs exhibit higher levels of genetic diversity compared to commercial pig breeds. In this study, a comprehensive analysis of the genetic diversity of indigenous pigs was conducted by investigating various factors that affect it. The results of the genetic diversity parameters used in this study, such as MAF, H_E_, H_O_, *pi*, LD and *F_ST_*, showed a significant level of genetic diversity in indigenous pigs. The significant artificial selection undergone by Western pig breeds over the past few decades has led to decreased genetic diversity in commercial pigs. These results are consistent with recent research. Wang et al. (2015), for example, reported that Chinese indigenous pigs exhibit greater genetic variability than commercial pigs [29]. Overall, the level of genetic diversity of these three indigenous pigs is relatively higher than commercial pig breeds, and the conservation effect has been positive in recent years. This will be advantageous in establishing future breeding strategies and selecting them.

In comparing the H_O_ and H_E_ between indigenous pigs and commercial pig breeds in this study, it was found that commercial pig breeds exhibit a higher value of H_E_. This inconsistency may be the reason of the SNP detection bias or the relatively small effective population size of indigenous pigs, coupled with limited sample representativeness, which has also been observed in many similar studies [30,31,32,33]. Interestingly, in comparing other indigenous pigs in China, such as the Jinhua pig or the Hongdenglong pig (with H_O_ values of 0.14 and 0.15, respectively) [34], our results suggest that the indigenous pig exhibits a higher genetic diversity. This could indicate that the indigenous pig has undergone less selection and suggests the strength of conservation efforts designed to protect it.

Pairwise Wright’s *F_ST_* is the common measure to estimate population genetic differentiation [31]. Based on our calculations, there is a substantial difference in *F_ST_* values between commercial pigs and native Chinese pigs, potentially due to geographic isolation. Therefore, in order to further investigate the differences between WZS and Hainan pigs, it is necessary to conduct further analysis of the population structure of Hainan local pigs.

### 4.2. Population Structure of Indigenous Pigs and Commercial Pigs

We performed genetic population structure analysis on five pig populations, and the patterns of the sub-clustering performed consistently among the analyses of NJ tree, PCA and STRUCTURE. The NJ tree displayed a clear bifurcation, similarly evident in the PCA plot. When K = 2 in the STRUCTURE analysis, we observed a marked division between local and commercial pigs in Hainan, consistent with the *F_ST_* values calculated between the two regions; this suggests that there are significant differences between local pigs and commercial pigs, possibly due to long-term genetic divergence and varying degrees of selection, as well as geographic isolation. It is worth noting that when K = 3 in structural analysis, TC and DA exhibit relatively similar characteristics, while WZS belongs to another group. Geographic isolation may also be responsible. The WZS pigs originated near the distant Wuzhishan City, whereas Tunchang City and Ding’an City are adjacent to each other—promoting crossbreeding between the TC and DA pigs. The mountainous terrain may have caused the WZS pig population to become isolated and independent from other populations. The distance between Tunchang and Ding’an is only 70 km. Considering that populations located in close geographic proximity are more likely to share common ancestors and mate with each other, locals believe that the DA pig may be the ancestor of the TC pig, and both populations are black pigs and have a similar appearance. This observation suggests that a closer genetic relationship between the two populations is possible, a hypothesis that has been supported by previous genetic studies. Based on these findings, it is clear that a more detailed analysis of the genetic and phenotypic traits of TC and DA is required in order to fully understand their evolutionary history and potential for economic development.

### 4.3. Gene Annotation and Functional Analysis

We analyzed three KEGG pathways, two of which are associated with intestinal immunity, while the other is linked to cholesterol metabolism. The pathways associated with intestinal immunity could involve genes or proteins related to inflammatory response, immune regulation and intestinal mucosal barrier, all of which play crucial roles in the production and immune response of IgA in the intestinal immune system.

Furthermore, the enrichment of the ssc04979, cholesterol metabolism pathway, may indicate that cholesterol metabolism plays a critical role in the physiological processes of indigenous pigs. This pathway involves multiple genes and molecules, including cholesterol synthesis and transport, bile acid synthesis and lipoprotein metabolism. The enrichment analysis of this pathway enables us to better comprehend the regulatory mechanism of cholesterol metabolism in Chinese indigenous pigs, providing us with new ideas and targets for improving the quality of pork and enhancing breeding efficiency.

We also identified several immune-related genes, such as *IL-2*, *IL-21* and *ZFYVE16*. Prior studies have reported *IL-2* as a growth factor that can drive the amplification of activated T cell populations in humans. *IL-21* is one of the primary immune modulators that regulates various immune responses by impacting different immune cells. T cells, as immune cells, play a crucial role in protecting the body against pathogens [35,36].

Additionally, *IL-13* belongs to the Th9 cytokine family and has significant immune regulatory activity that affects various immune cells. *IL-17A* promotes the production of corresponding myeloid cells by recruiting pathogenic T cells and plays an initiating role in autoimmunity [37]. Moreover, *ZFYVE16* may upregulate the proliferation of B lymphocytes through TGF-β signal transduction [38]. The combined action of these genes may explain the high levels of disease tolerance associated with the indigenous pigs of Hainan.

As there are survival differences between commercial pigs and local pigs, we have identified genes that may be related to adaptability, such as *APBA1*. Previous research has linked *APBA1* to cognitive function, and its absence has been associated with the onset of neurobehavioral disorders [39,40]. This could indicate that Chinese indigenous pigs are more intelligent and better suited to adapting to their environment.

We have also identified several genes related to reproduction. The *FGF2* gene mediates *FGF2/FGFR* signal transduction, which can promote in vitro maturation of cumulus-oocyte complexes. *RNF17* is a component of mammalian germ cells and is critical for sperm development [41,42]. *ADAD1* is a protein necessary for male fertility that contains the testis-specific adenosine deaminase domain. Additionally, *HIPK4* is essential for mouse sperm development [43,44]. It is especially noteworthy that despite living in a hot and humid environment, Chinese indigenous pigs still exhibit good reproductive ability. This suggests that the expression of these genes may reflect high levels of adaptation to the environmental conditions in which the pigs live, allowing them to maintain their reproductive fitness. Similarly, our results may explain why Chinese indigenous pigs exhibit strong sperm vitality and are considered highly tolerant of their environment. Furthermore, understanding the genetic basis of fertility and sperm development in Chinese indigenous pigs could aid in improving breeding programs and enhancing productivity. By identifying genes linked to reproductive performance, we can select pigs with desirable traits and improve their overall breeding efficiency.

Moreover, some candidate genes, such as *ABCA1*, *ADIG*, *TLE4* and *IRX5*, have been found to be associated with lipid metabolism and fat deposition. Intramuscular fat (IMF) is considered a major factor that affects meat quality. Often, genes related to lipid metabolism, such as *ABCA1*, are downregulated in commercial meats, such as Wenchang chicken [45]. The selection signal we observed in ABCA1 could be related to the fact that Chinese indigenous pigs exhibit better meat quality. In a similar vein, research indicates that ADIG plays a role in regulating fat accumulation and leptin secretion in mice. Additionally, studies have shown that this gene is associated with muscle development mechanisms in Yunling cattle [46,47]. *TLE4* may regulate the quiescence of muscle stem cells and influence skeletal muscle differentiation. *IRX5* promotes the generation of adipose tissue in hMSCs by inhibiting glycolysis, which may further explain the delicious taste associated with the indigenous pigs [48,49]. Furthermore, understanding the genetic basis of lipid metabolism and fat deposition in Chinese indigenous pigs could aid in improving meat quality and enhancing breeding programs. By identifying genes related to lipid metabolism, we can select pigs with desirable traits and improve their overall breeding efficiency.

It is worth noting that Asadollahi et al. (2022) suggested that *VPS13A*, which encodes the protein Chorein, may play a significant role in heat stress resistance. Moreover, Schmidt et al. (2013) identified *VPS13A* as a critical regulator of secretion and blood platelet aggregation during heat stress conditions. This emphasizes the importance of comprehending the molecular mechanisms that underlie the response to heat stress in different biological scenarios at both cellular and organismal levels [50,51]. In addition, heat shock protein A4 (*HSPA4*), a member of the HSP110 family, can control apoptosis of inflammatory cells and immune reactions in the gut [52]. In Liu et al.’s study, it was found that the expression of *Hsp110* in primary cardiomyocytes in vitro was sensitive to heat stress, and *Hsp110* participated in the potential acquisition of heat tolerance after heat stress [53]. Therefore, these genes may be related to heat tolerance in Chinese local pigs, providing guidance for further exploration of heat tolerance genes.

Several QTLs have been identified that are linked to meat quality traits, including drip loss, loin muscle area, muscle moisture percentage and shoulder subcutaneous fat thickness. Chinese indigenous pigs are renowned for their exceptional meat quality, and the overlapping of QTLs with potential selection regions may provide an explanation for the genetic differences observed between Chinese indigenous pig populations and commercial pig breeds. This information could potentially aid in the development of breeding programs that aim to improve the meat quality of commercial pig breeds.

In summary, these findings are of great significance for understanding the unique characteristics of Chinese local pig populations, which will be beneficial for our understanding of Chinese local pigs as a whole. Moreover, the insights gained from this study provide a foundation for proposing a reliable and sustainable strategy for the conservation and improvement of indigenous pigs. This information can be utilized to optimize breeding programs, enhance genetic diversity and promote the preservation of local pig populations, ultimately contributing to the overall biodiversity of livestock species.

## 5. Conclusions

Knowledge of the genetic diversity and population structure for one population may provide valuable information for its effective conservation and utilization. We found that the indigenous pigs tested in this study are much more genetically diverse than commercial pigs. Furthermore, we identified pathways and genes related to meat quality, immunity and adaptability in the indigenous pigs—valuable genetic resources that must be conserved for years to come. Overall, we conclude that conservation of indigenous pigs has been effective to date, but their population protection plan still needs further optimization to avoid inbreeding depression and to maintain sufficient genetic diversity.

## Figures and Tables

**Figure 1 animals-13-02010-f001:**
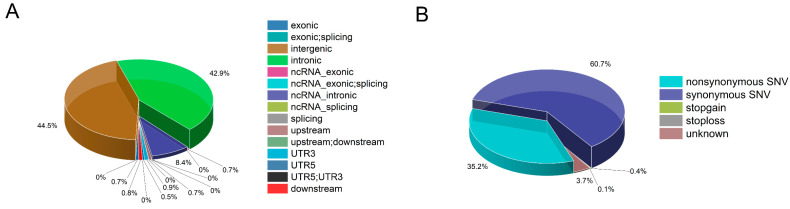
SNP characteristics of indigenous pig population. (**A**) This figure shows the annotation of all variations within the three indigenous pig populations, with each variation area annotated as a percentage. (**B**) This figure displays the annotated variation information within the coding region of the three indigenous pig populations.

**Figure 2 animals-13-02010-f002:**
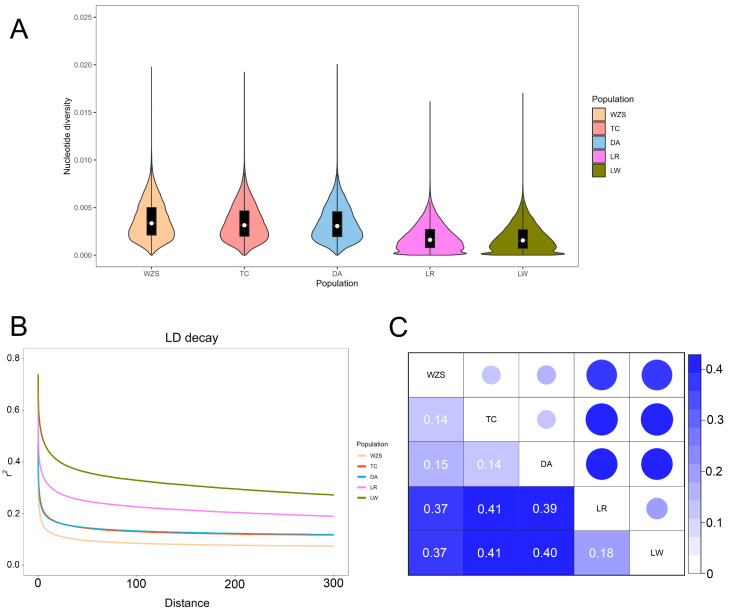
Genetic diversity of indigenous pigs and commercial pigs. (**A**) Drawn violin plots for nucleotide diversity in each population. (**B**) Drawn LD decay plots for each population. This plot describes the change in the degree of linkage disequilibrium (LD) between two loci along the distance. (**C**) Heatmap of *F_ST_* distance between populations.

**Figure 3 animals-13-02010-f003:**
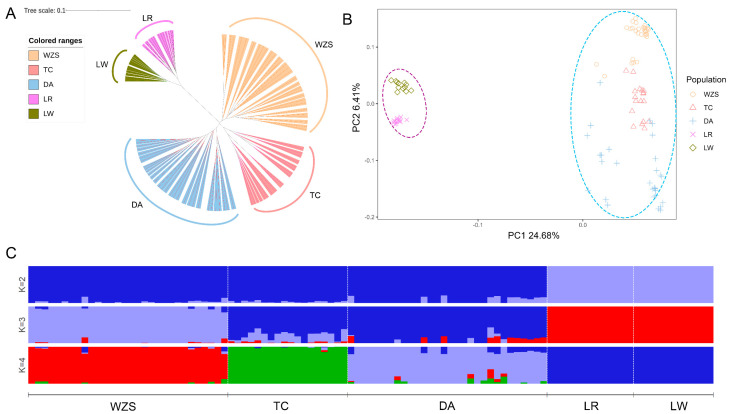
Analysis of genetic structure of the tested pigs. WZS, Wuzhishan pigs; TC, Tunchang pigs; DA, Dingan pigs; LR, Landrace pigs; LW, Large White pigs. (**A**) Neighbor-joining tree for all individuals. (**B**) Plot of the first and second principal components resulting from a principal component analysis of all pig populations. (**C**) The plot of population structure for all pig populations (K = 2–4). Different colors represent different clusters.

**Figure 4 animals-13-02010-f004:**
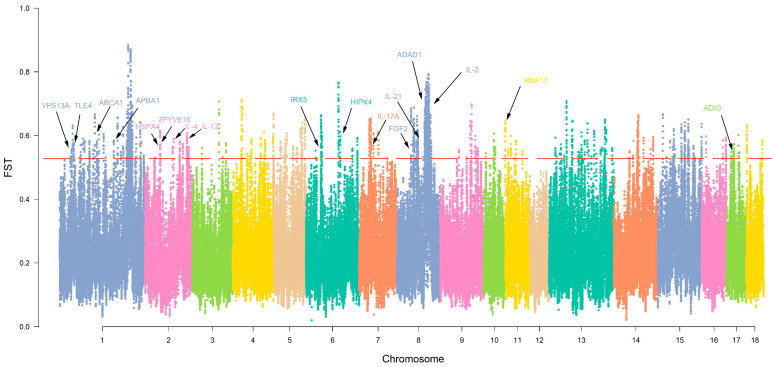
Manhattan plot of selective signatures by *F_ST_* in the Chinese indigenous pigs. The red dotted line means the threshold for classifying outliers in the heat group (top 1%). Different colors are used to distinguish the neighboring chromosomes. Several related candidate genes are also highlighted.

**Figure 5 animals-13-02010-f005:**
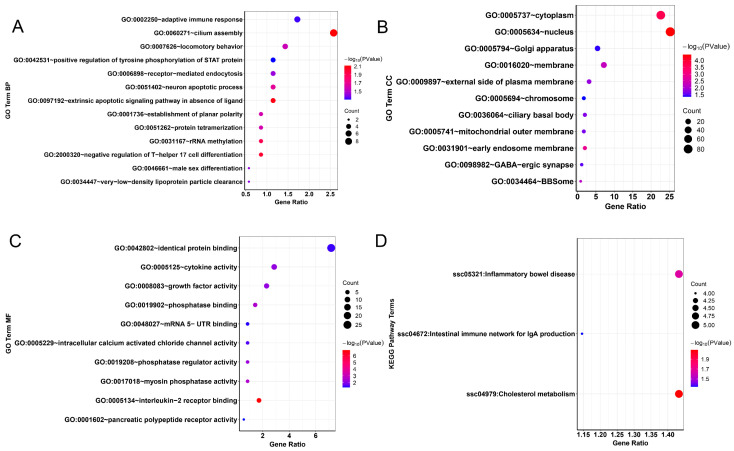
GO terms and KEGG pathways were drawn for the candidate exonic genes that were screened based on *F_ST_*. (**A**) Bubble chart illustrating Gene Ontology (GO) Biological Process enrichment analysis results. (**B**) Bubble chart demonstrating Gene Ontology (GO) Cellular Component enrichment analysis findings. (**C**) Bubble chart highlighting Gene Ontology (GO) Molecular Function enrichment analysis outcomes. (**D**) Bubble chart illustrating KEGG pathway analysis results of selected candidate genes.

**Table 1 animals-13-02010-t001:** Genetic diversity of Chinese indigenous pigs and commercial pigs.

Population	Abbreviation	Size	MAF	H_O_	H_E_	Nucleotide Diversity (*pi*)
Wuzhishan	WZS	30	0.22 ± 0.13	0.30 ± 0.15	0.31 ± 0.14	0.0037
Tunchang	TC	18	0.23 ± 0.13	0.33 ± 0.18	0.32 ± 0.14	0.0034
Dingan	DA	30	0.23 ± 0.14	0.34 ± 0.17	0.32 ± 0.14	0.0034
Landrace	LR	13	0.24 ± 0.13	0.29 ± 0.15	0.34 ± 0.12	0.0019
Large White	LW	12	0.26 ± 0.13	0.39 ± 0.18	0.35 ± 0.12	0.0018

## Data Availability

The raw data used in this study are publicly available and can be obtained upon reasonable request to the corresponding author.

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
