# Peer review of "Evaluation of the Genetic Diversity, Population Structure and Selection Signatures of Three Native Chinese Pig Populations"

_animals, 2023, doi:10.3390/ani13122010_

Round 1

Reviewer 1 Report

The authors analyse whole-genome sequencing data of 78 animals from three Hainan pig populations (Wuzhishan, Tunchang, and Ding'an) and WGS data of 25 foreign pigs (Landrace and Large White). Each of the Hainan pig population represented a unique genetic resource. The selection signatures detected between Hainan and foreign pig populations identified genes which may affect immunity, adaptability, reproduction, meat quality, and heat tolerance.

Introduction: the authors may summarize previous results on pig breeds obtained from NGS data in a supplementary Table. It seems not necessary to mention other species than pigs here. There are numerous papers. Please also refer to the methods you are applying.

No comments.

Author Response

Response: Thank you for your suggestions. We have added a supplementary table in the Materials and Methods section and limited the discussion in the Introduction to pigs only, without mentioning other species. Additionally, we have included several new references on pig research.

Reviewer 2 Report

This manuscript brings important genetic information on Chinese indigenous pig populations that will contribute to the effective conservation of these genetic resources.

Suggestions or corrections in the manuscript:

I suggest changing the title to something more direct and informative: “Evaluation of the genetic diversity, population structure and selection signatures of three native Chinese pig populations”.

Throughout the manuscript, "in Hainan Province" is repeatedly inserted: thinking of a more internationalized writing, I suggest revising the entire text and leaving it exclusively in the Material and Methods or when this specificity of the region within China is essential.

Use "Keywords" that are not in the title.

Line 112 it is important to descriptively describe the name of the three breeds of pigs "Wuzhishan (WZS), Tunchang (TC) and Dingan (DA)", considering that the reader will naturally look for this information in Materials and Methods.

Figure 1: part B of the figure the values 0.1% and 0.4% it is not possible to identify the color and know if it is stopgain or stoploss. Need to fix.

Line 217 correct: "from 014" to "from 0.14".

Figure 5 - I suggest making the graphics bigger, even if they can't be next to each other, but they will be visible.

Line 432-435 withdraw: "In summary, our study analyzed the genetic diversity, population structure, and conservation evaluation of three local pig populations in Hainan and compared them to two common commercial pigs. These findings are of great significance for understanding the unique". Insert: "In summary, these findings are of great significance for understanding the unique".

Author Response

Point 1: I suggest changing the title to something more direct and informative: “Evaluation of the genetic diversity, population structure and selection signatures of three native Chinese pig populations”.

Response 1: Thank you for your suggestion. We have changed the title to“Evaluation of the genetic diversity, population structure and selection signatures of three native Chinese pig populations”.

Point 2: Throughout the manuscript, "in Hainan Province" is repeatedly inserted: thinking of a more internationalized writing, I suggest revising the entire text and leaving it exclusively in the Material and Methods or when this specificity of the region within China is essential.

Response 2: Many thanks for your comments. We have removed excessive mentions of Hainan Province in the article, in order to make it more nationalistic.

Point 3: Use "Keywords" that are not in the title.

Response 3: Thank you for your input. We have revised the keywords as follows: Chinese indigenous pigs; single nucleotide polymorphism; genetic variation; conservation; and F-statistic.

Point 4: Line 112 it is important to descriptively describe the name of the three breeds of pigs "Wuzhishan (WZS), Tunchang (TC) and Dingan (DA)", considering that the reader will naturally look for this information in Materials and Methods.

Response 4: Many thanks for your comments. We have made modifications to the abbreviation in this sentence, see Line103-104.

Point 5: Figure 1: part B of the figure the values 0.1% and 0.4% it is not possible to identify the color and know if it is stopgain or stoploss. Need to fix.

Response 5: Many thanks for your comments. We have made modifications with this figure to make the information clearer.

Point 6: Line 217 correct: "from 014" to "from 0.14".

Response 6: Thank you for your suggestion. This was our negligence and we will change the word to 'from 0.14', see Line 234.

Point 7: Figure 5 - I suggest making the graphics bigger, even if they can't be next to each other, but they will be visible.

Response 7: Thank you for your suggestion. We have adjusted the graphic font to make the image more clearer.

Point 8: Line 432-435 withdraw: "In summary, our study analyzed the genetic diversity, population structure, and conservation evaluation of three local pig populations in Hainan and compared them to two common commercial pigs. These findings are of great significance for understanding the unique". Insert: "In summary, these findings are of great significance for understanding the unique".

Response 8: Thank you for your suggestion. We have changed the sentence to " In summary, these findings are of great significance for understanding the unique characteristics of Hainan's local pig populations, which will be beneficial for our understanding of Chinese local pigs as a whole", see Line 456-457.

Reviewer 3 Report

The manuscript was well written, and the data was collected using structured method which advance on its field currently.

Minor corrections were as follows:

Line 118: method for library construction need to be adeed

Line 145: all the bioinformatic database and tools must be accompanied with the website link for it source

Line 240: ind pig ? please explain

Line 250: what was the reason taking 1% of the right-tail?, give a brief explanation here

Line 252: add the link for this bioinformatic resource

Line 281: please clarify this Figure 5 tittle in the body text. See line 255-256. Add explanation that the GO and KEEG pathway analysis was based on the known gene exon.

English was fine, and understandable. Minor grammatical error need to be fixed but not much to found.

Author Response

Point 1: Line 118: method for library construction need to be adeed

Response 1: Thank you for your suggestion. We have described the method of constructing the library, see Line 119-127.

Point 2: Line 145: all the bioinformatic database and tools must be accompanied with the website link for it source

Response 2: Thank you for your suggestion. We have included the website for downloading the software in the article, see Line 143-157.

Point 3: Line 240: ind pig ? please explain

Response 3: Thank you for your suggestion. It was our oversight, we will change the word to 'indigenous pigs', see Line 209.

Point 4: Line 250: what was the reason taking 1% of the right-tail?, give a brief explanation here

Response 4: Thank you for considering my suggestion. The revised statement would be "Screening candidate genes using a 1% FST value for statistics " which is based on the studies "Identification of Signatures of Selection by Whole-Genome Resequencing of a Chinese Native Pig" and "A genome-wide scan for signatures of selection in Chinese indigenous and commercial pig breeds". We also used the top 1% values for selection screening, see Line 267.

Point 5: Line 252: add the link for this bioinformatic resource

Response 5: Thank you for your suggestion. We have added the website address in this line, please check line 267.

Point 6: Line 281: please clarify this Figure 5 tittle in the body text. See line 255-256. Add explanation that the GO and KEEG pathway analysis was based on the known gene exon.

Response 6: Thank you for your feedback. We have added the sentence 'GO terms and KEGG pathway analyses are based on known genes and their exons' to make the statement more precise.
